# TRAINING GROUP ORTHOGONAL NEURAL NETWORKS WITH PRIVILEGED INFORMATION

**Yunpeng Chen[1], Xiaojie Jin[1], Jiashi Feng[1], Shuicheng Yan[2,1]**

[1]National University of Singapore

[2]Qihoo 360 AI Institute

{chenyunpeng,xiaojie.jin}@u.nus.edu

elefjia@nus.edu.sg

yanshuicheng@360.cn

## ABSTRACT

Learning rich and diverse feature representation are always desired for deep convolutional neural networks (CNNs). Besides, when auxiliary annotations are available for specific data, simply ignoring them would be a great waste. In this paper, we incorporate these auxiliary annotations as privileged information and propose a novel CNN model that is able to maximize inherent diversity of a CNN model such that the model can learn better feature representation with a stronger generalization ability. More specifically, we propose a group orthogonal convolutional neural network (GoCNN) to learn features from foreground and background in an orthogonal way by exploiting privileged information for optimization, which automatically emphasizes feature diversity within a single model. Experiments on two benchmark datasets, ImageNet and PASCAL VOC, well demonstrate the effectiveness and high generalization ability of our proposed GoCNN models.

## 1 INTRODUCTION

Deep convolutional neural networks (CNNs) have brought a series of breakthroughs in image classification tasks (He et al., 2015; Girshick, 2015; Zheng et al., 2015). Many recent works (Simonyan & Zisserman, 2014; He et al., 2015; Krizhevsky et al., 2012) have observed that CNNs with different architectures or even different weight initializations may learn slightly different feature representations. Combining these heterogeneous models can provide richer and more diverse feature representation which can further boost the final performance. Such observation motivate us to directly pursue feature diversity within a single model in the work.

Besides, many existing datasets (Everingham et al., 2010; Deng et al., 2009; Xiao et al., 2010) provide more than one types of annotations. For example, the PASCAL VOC (Everingham et al., 2010) provides image level tags, object bounding box, and image segmentation masks; the ImageNet dataset (Deng et al., 2009) provide image level tags and a small portion of bounding box. Only using the image level tags for training image classification model would be a great waste on the other annotation resources. Therefore, in this work, we investigate whether these auxiliary annotations could also help a CNN model learn richer and more diverse feature representation.

In particular, we take advantage of these extra annotated information during training a CNN model for obtaining a *single* CNN model with sufficient inherent diversity, with the expectation that the model is able to learn more diverse feature representations and offers stronger generalization ability for image classification than vanilla CNNs. We therefore propose a group orthogonal convolutional neural network (GoCNN) model that is able to exploit these extra annotated information as privileged information. The idea is to learn different groups of convolutional functions which are "orthogonal" to the ones in other groups. Here by "orthogonal", we mean there is no significant correlation among the produced features. By "privileged information", we mean these auxiliary information only been used during the training phase. Optimizing orthogonality among convolutional functions reduces redundancy and increases diversity within the architecture.

Properly defining the groups of convolutional functions in the GoCNN is not an easy task. In this work, we propose to exploit available *privileged information* for identifying the proper groups. Specifically, in the context of image classification, object segmentation annotations which are (partially) available in several public datasets give richer information.

In addition, the background contents are usually independent on foreground objects within an image. Thus, splitting convolutional functions into different groups and enforcing them to learn features from the foreground and background separately can help construct orthogonal groups with small correlations. Motivated by this, we introduce the GoCNN architecture which explores to learn discriminative features from foreground and background separately where the foreground-background segregation is offered by the privileged segmentation annotation for training GoCNN. In this way, inherent diversity of the GoCNN can be explicitly enhanced. Moreover, benefiting from pursuing the group orthogonality, the learned convolutional functions within GoCNN are demonstrated to be foreground and background diagnostic even for extracting features from new images in the testing phase.

To the best of our knowledge, this work is the first to explore a principled way to train a deep neural network with desired inherent diversity and the first to investigate how to use the segmentation privileged information to assist image classification within a deep learning architecture. Experiments on ImageNet and PASCAL VOC clearly demonstrate GoCNN improves upon vanilla CNN models significantly, in terms of classification accuracy.

As a by-product of implementing GoCNN, we also provide positive answers to the following two prominent questions about image classification: (1) Does background information indeed help object recognition in deep learning? (2) Can a more precise annotation with richer information, *e.g.*, segmentation annotation, assist the image classification training process non-trivially?

## 2 RELATED WORK

Learning rich and diverse feature representations is always desired while training CNNs for gaining stronger generalization ability. However, most existing works mainly focus on introducing hand-crafted cost functions to implicitly pursue diversity (Tang, 2013), or modifying activation functions to increase model non-linearity (Jin et al., 2015) or constructing a more complex CNN architecture (Simonyan & Zisserman, 2014; He et al., 2015; Krizhevsky et al., 2012). Methods that explicitly encourage inherent diversity of CNN models are still rare so far.

Knowledge distillation (Hinton et al., 2015) can be seen as an effective way to learn more discriminative and diverse feature representations. The distillation process compresses knowledge and thus encourages a weak model to learn more diverse and discriminative features. However, knowledge distillation works in two stages which are isolated from each other and has to rely on pre-training a complicated teacher network model. This may introduce undesired computation overhead. In contrast, our proposed approach can learn a diverse network in a *single* stage without requiring an extra network model. Similar works, *e.g.* the Diversity Networks (Sra & Hosseini), also squeeze the knowledge by preserving the most diverse features to avoid harming the performance.

More recently, Cogswell et al. (2016) proposed the DeCov approach to reduce over-fitting risk of a deep neural network model by reducing feature covariance. DeCov also agrees with increasing generalization ability of a model by pursuing feature diversity. This is consistent with our motivation. However, DeCov penalizes the covariance in an unsupervised fashion and cannot utilize extra available annotations, leading to insignificant performance improvement over vanilla models (Cogswell et al., 2016).

Using privileged information to learn better features during the training process is similar in spirit with our method. Both our proposed method and Lapin et al. (2014) introduce privileged information to assist the training process. However, almost all existing works (Lapin et al., 2014; Lopez-Paz et al., 2016; Sharmanska et al., 2014) are based on $SVM^+$ which only focuses on training a better classifier and is not able to do the end-to-end training for better features.

Several works (Andrew et al., 2013; Srivastava & Salakhutdinov, 2012) about canonical correlation analysis (CCA) for CNNs provide a way to constrain feature diversity. However, the goal of CCA

is to find linear projections for two random vectors that are maximally correlated, which is different from ours.

It is also worth to notice that simply adding a segmentation loss to image classification neural network is not equivalent to a GoCNN model. This is because image segmentation requires each pixel within the target area to be activated and the others stay silent for dense prediction, while GoCNN does not require the each pixel within the target area to be activated. GoCNN is specifically designed for classification tasks, not for segmentation ones. Moreover, our proposed GoCNN supports learning from partial privileged information wile the CNN above needs a fully annotated training set.

## 3 MODEL DIVERSITY OF CONVOLUTIONAL NEURAL NETWORKS

Throughout the paper, we use $f(\cdot)$ to denote a convolutional function (or filter) and $k$ to index the layers in a multi-layer network. We use $c^{(k)}$ to denote the total number of convolutional functions at the $k$-th layer and use $i$ and $j$ to index different functions, *i.e.*, $f_i^{(k)}(\cdot)$ denotes the $i$-th convolutional function at the $k$-th layer of the network. The function $f$ maps an input feature map to another new feature map. The height and the width of a feature map output at the layer $k$ are denoted as $h^{(k)}$ and $w^{(k)}$ respectively. We consider a network model consisting of $N$ layers in total.

Under a standard CNN architecture, the elements within the same feature map are produced by the same convolutional function $f_i^{(k)}$ and thus they represent the same type of features across different locations. Therefore, encouraging the feature variance or diversity within a single feature map does not make sense. In this work, our target is to enhance the diversity among different convolutional functions. Here we first give a formal description of *model diversity* for an $N$-layer CNN.

**Definition 1** (Model Diversity). *Let $f_i^{(k)}$ denote the $i$-th convolutional function at the $k$-th layer of a neural network model, and then the model diversity of the $k$-th layer is defined as*

$$\zeta^{(k)} \triangleq 1 - \frac{1}{c^{(k)^2}} \sum_{i,j=1}^{c^{(k)}} \mathrm{cor}\left(f_i^{(k)}, f_j^{(k)}\right). \tag{1}$$

*Here the operator $\mathrm{cor}(\cdot, \cdot)$ denotes the statistical correlation.*

In other words, the inherent diversity of a network model that we are going to maximize is evaluated across all the convolutional functions within the same layer.

The most straightforward way to maximize the above diversity for each layer is to directly maximize the quantity of $\zeta^{(k)}$ during training the network. However, it is quite involved to optimize the hard diversity in (1) due to the large combination number of different convolutional functions. Thus, we propose to solve this problem by learning the convolutional functions in different *groups* separately. Different functions from different groups are uncorrelated to each other and we do not need to consider their correlation. Suppose the convolutional functions at each layer are partitioned into $m$ different groups, denoted as $\mathcal{G} = \{G_1, \ldots, G_m\}$. Then, we instead maximize the following *Group-wise Model Diversity*.

**Definition 2** (Group-wise Model Diversity). *Given a pre-defined group partition set $\mathcal{G} = \{G_1, \ldots, G_m\}$ of convolutional functions at a specific layer, the group-wise model diversity of this layer is defined as*

$$\zeta_g^{(k)} \triangleq 1 - \frac{1}{c^{(k)^2}} \sum_{s,t=1}^{|\mathcal{G}|} \sum_{i \in G_s, j \in G_t} \mathrm{cor}\left(f_i^{(k)}, f_j^{(k)}\right).$$

Instead of directly optimizing the *model diversity*, we consider optimizing the *group-wise model diversity* by finding a set of orthogonal groups $\{G_1^*, \ldots, G_m^*\}$, where convolutional functions within each group are uncorrelated with others within different groups. In the scenario of image representation learning, one typical example of such orthogonal groups is the foreground group and background group pair — partitioning the functions into two groups and letting them learn features from foreground and background contents respectively.

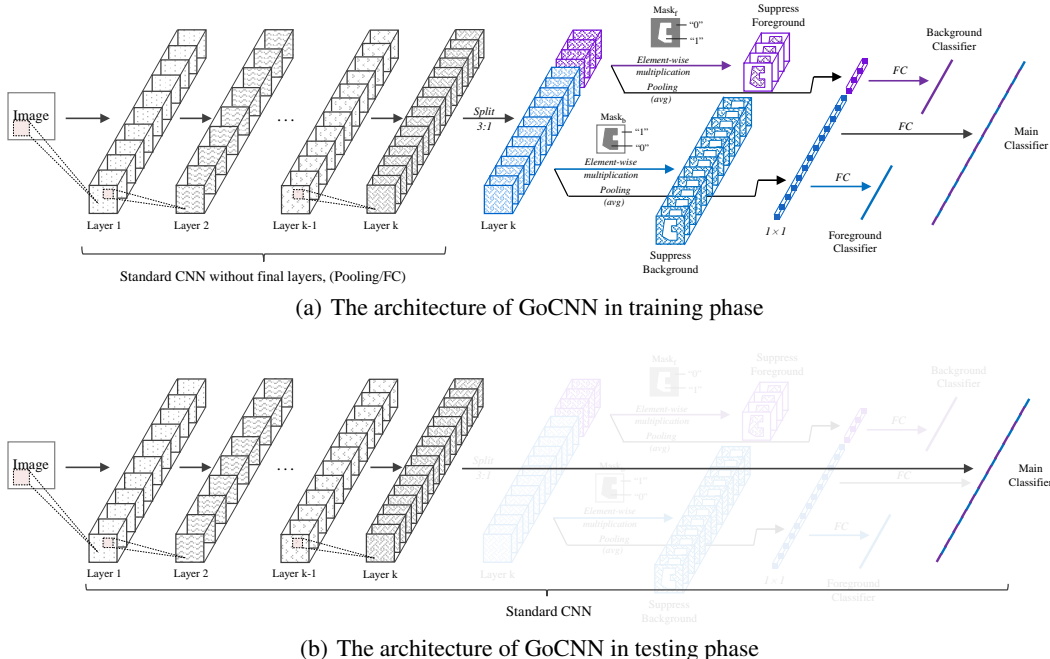

(a) The architecture of GoCNN in training phase

(b) The architecture of GoCNN in testing phase

Figure 1: Architectures of the proposed GoCNN used in the training (top) and testing (bottom) phase. The two groups are colored by blue (foreground) and purple (background) respecively. FC represents the fully connected layer.

In this work, we use segmentation annotation as privileged information for finding orthogonal groups of convolutional functions $G_1^*, \ldots, G_m^*$. In particular, we derive the foreground and background segregation from the privileged information for an image. Then we partition convolutional functions at a specific layer of a CNN model into *foreground* and *background* groups respectively, and train a GoCNN model to learn the foreground and background features separately. Details about the architecture of the GoCNN and the training procedure of GoCNN are given in the following section.

## 4 GROUP ORTHOGONAL CONVOLUTIONAL NEURAL NETWORKS

We introduce the group orthogonal constraint to maximize group-wise diversity among different groups of convolutional functions explicitly by constructing a group orthogonal convolutional neural network (GoCNN). Details on the architecture of GoCNN are shown in Figure 1. GoCNN is built upon a standard CNN architecture. The convolutional functions at the final convolution layer are explicitly divided into two groups: the foreground group which concentrates on learning the foreground feature and the background group which learns the background feature. The output features of these two groups are then aggregated by a fully connected layer.

In the following subsections, we give more details of the foreground and background groups construction. After that, we will describe how to combine these two components and build them into a unified network architecture — the GoGNN.

### 4.1 FOREGROUND AND BACKGROUND GROUPS

To learn convolutional functions that are specific for foreground content of an image, we propose the following two constraints for the foreground group of functions. The first constraint forces the functions to be learned from the foreground only and free of any contamination from the background, and the second constraint encourages the learned functions to be discriminative for image classification.

We learn features that only lie in the foreground by suppressing any contamination from the background. As aforementioned, here we use the object segmentation annotations (denoted as $\mathrm{Mask}$) as the privileged information in the training phase to help identify the background features where the foreground convolutional functions should not respond to. The background contamination is extracted by an extractor adopted on each feature map within the foreground group. In particular, we define an extractor $\varphi(\cdot, \cdot)$ as follows:

$$\varphi(f_i^{(k)}(x), \mathrm{Mask}) \triangleq f_i^{(k)}(x) \odot \mathrm{Mask}, \tag{2}$$

where $x$ denotes the raw input and $\odot$ denotes the element-wise multiplication.

In the above operator, we use the background object mask $\mathrm{Mask}_b$ to extract background features. Each element in $\mathrm{Mask}_b$ is equal to one if the corresponding position lies on a background object or zero otherwise. Here, we assume the masks are already re-sized to have compatible dimensionality with the output feature map $f_i^{(k)}(x)$ by the interpolation method so that the element-wise multiplication is valid here. The extracted background features are then suppressed by a regression loss defined as follows:

$$\min_\theta \sum_i \|\varphi(f_i^{(k)}(x; \theta), \mathrm{Mask}_b)\|_F. \tag{3}$$

Here $\theta$ parameterizes the convolution function $f_i^{(k)}$. Since the target value for this regression is zero, we also call it a *suppression* term. It will only suppress the response output by $f_i^{(k)}$ at the locations outside the foreground.

For the second constraint, *i.e.*, encouraging the functions to learn discriminative features, we simply use the standard softmax classification loss to supervise the learning phase.

The role of the background group is complementary to the foreground one. It aims to learn convolutional functions that are only specific for background contents. Thus, the functions within the background group have a same suppression term as in Eqn. (3), in which $\mathrm{Mask}_b$ is replaced with $\mathrm{Mask}_f$ to restrict the learned features to make them only lie in the background space. The $\mathrm{Mask}_f$ is simply computed as $\mathrm{Mask}_f = 1 - \mathrm{Mask}_b$. Also, a softmax linear classifier is attached during training to guarantee that these learned background functions are useful for predicting image categories.

## 4.2 ARCHITECTURE AND IMPLEMENTATION DETAILS OF THE GOCNN

In GoCNN, the size ratio of foreground group and background group is fixed to be 3:1 during training, since intuitively the foreground contents are much more informative than the background contents in classifying images. A single fully connected layer (or multiple layers depending on the basic CNN architecture) is used to unify the functional learning within different groups and combine features learned from different groups. It aggregates the information from different feature spaces and produces the final image category prediction. More details are given in Figure 1.

Because we are dealing with the classification problem, a main classifier with a standard classification loss function is adopted at the top layer of GoCNN. In our experiments, the standard softmax loss is used for single-label image classification and the logistic regression loss is used for multiple-label image classification, *e.g.*, images from the Pascal VOC dataset (Everingham et al., 2010).

During the testing stage, parts unrelated to the final main output will be removed, as shown in Figure 1 (b). Therefore, in terms of testing, neither extra parameters nor extra computational cost is introduced. The GoCNN is exactly the same as the adopted CNN in the testing phase.

In summary, for an incoming training sample, it passes through all the layers to the final convolution layer. Then the irrelevant features for each group (foreground or background) will be filtered out by privileged segmentation masks. Those filtered features will then flow into a suppressor (see Eqn. (3)). For the output features from each group, it will flow up along two paths: one leads to the group-wise classifier, and the other one leads to the main classifier. The three gradients from the suppressors, the group-wise classifiers and the main classifier will be used for updating the network parameters.

**Applications with Incomplete Privileged Information** Our proposed GoCNN can also be applied for semi-supervised learning. When only a small subset of images have the privileged seg-

mentation annotations in a dataset, we simply set the segmentations of images without annotations to be $\text{Mask}_f = \text{Mask}_b = \mathbf{1}$, where $\mathbf{1}$ is the matrix with all of its elements being 1. In other words, we disable both the suppression terms (ref. Eqn. (3)) on foreground and background parts as well as the extractors on the back propagation path. By doing so, fully annotated training samples with privileged information will supervise GoCNN to learn both discriminative and diverse features while the samples with only image tags only guide GoCNN to learn category discriminative features.

## 5 EXPERIMENTS

### 5.1 EXPERIMENT SETTINGS AND IMPLEMENTATION DETAILS

**Datasets**   We evaluate the performance of GoCNN in image classification on two benchmark datasets, *i.e.*, the ImageNet (Deng et al., 2009) dataset and the Pascal VOC 2012 dataset (Everingham et al., 2010).

- **ImageNet**   ImageNet contains 1,000 fine-grained classes with about 1,300 images for each class and 1.2 million images in total, but without any image segmentation annotations. To collect privileged information, we randomly select 130 images from each class and manually annotate the object segmentation masks for them. Since our focus is on justifying the effectiveness of our proposed method, rather than pushing the state-of-the-art, we only collect privileged information for 10% data (overall 130k training images) to show performance improvement brought by our model. We call the new dataset consisting of these segmented images as *ImageNet-0.1m*. For evaluation, we use the original validation set of ImageNet which contains 50,000 images. Note that neither our baselines nor the proposed GoCNN needs segmentation information in testing phase.
- **PASCAL VOC 2012**   The PASCAL VOC 2012 dataset contains 11,530 images from 20 classes. For the classification task, there are 5,717 images for training and 5,823 images for validation. We use this dataset to further evaluate the generalization ability of different models including GoCNN trained on the ImageNet-0.1m: we pre-train the evaluated models on the ImageNet-0.1m dataset and fine-tune them using the logistic regression loss on PASCAL VOC 2012 training set. We evaluate their performance on the validation set.

**The Basic Architecture of GoCNN**   In our experiments, we use the ResNet (He et al., 2015) as the basic architecture to build GoCNN. Since the deepest ResNet contains 152 layers which will cost several weeks to train, we choose a light version of architecture (ResNet-18 (He et al., 2015)) that contains 18 layers as our basic model for most cases. We also use the ResNet-152 (He et al., 2015) for experiments on the full ImageNet dataset. The final convolution layer gives a $7 \times 7$ output and is pooled into a $1 \times 1$ feature map by average pooling. Then a fully connected layer is added to perform linear classification. The used loss function for the single class classification on ImageNet dataset is the standard softmax loss. When performing multi-label classification on *PASCAL VOC*, we use the logistic regression loss.

**Training and Testing Strategy**   We use MXNet (Chen et al., 2015) to conduct model training and testing. The GoCNN weights are initialized as in He et al. (2015) and we train GoCNN from scratch. Images are resized with a shorter side randomly sampled within [256, 480] for scale augmentation and $224 \times 224$ crops are randomly sampled during training (He et al., 2015). We use SGD with base learning rate equal to 0.1 at the beginning and reduce the learning rate by a factor of 10 when the validation accuracy saturates. For the experiments on ResNet-18 we use single node with a mini-batch size of 512. For the ResNet-152 we use 48 GPUs with mini-batch size of 32 for each GPU. Following He et al. (2015), we use a weight decay of 0.0001 and a momentum of 0.9 in the training.

We evaluate the performance of GoCNN on two different testing settings: the complete privileged information setting and the partial privileged information setting. We perform 10-crop testing (Krizhevsky et al., 2012) for the complete privileged information scenario, and do a single crop testing for the partial privileged information scenario for convenience.

**Compared Baseline Models**   Our proposed GoCNN follows the Learning Using Privileged Information (LUPI) paradigm (Lapin et al., 2014), which exploits additional information to facilitate

Table 1: Validation accuracy (for 10-crop validation) of different models on ImageNet validation set. All the compared models are trained on the ImageNet-0.1m dataset with complete privileged information.

|  | Top-1 Accuracy (%) | | | Top-5 Accuracy (%) | | |
|---|---|---|---|---|---|---|
|  | Main_classifier | Fg_classifier | Bg_classifier | Main_classifier | Fg_Classifier | Bg_Classifier |
| SVM+ | 37.53 | — | — | — | — | — |
| Baseline | 46.00 | — | — | 70.05 | — | — |
| Full GoCNN | **50.39** | 49.60 | 40.03 | **75.00** | 74.21 | 66.98 |

learning but does not require extra information in testing. There are a few baseline models falling into the same paradigm that we can compare with. One is the SVM+ method (Pechyony & Vapnik, 2011) and the other one is the standard model (*i.e.*, the ResNet-18). We simply refer to ResNet-18 by *baseline* if no confusion occurs. In the experiments, we implement the SVM+ using the code provided by Pechyony & Vapnik (2011) with default parameter settings and linear kernel. We follow the scheme as described in Lapin et al. (2014) to train the SVM+ model. More concretely, we train multiple one-versus-rest SVM+ models upon the deep features extracted from both the entire images and the foreground regions (used as the privileged information). We use the averaged pooling over 10 crops on the feature maps before the $FC$ layer as the deep feature for training SVM+. It is worth noting that all of these models (including SVM+ and GoCNN) use a linear classifier and thus have the same number of parameters, or more concretely, GoCNN does not require more parameters than SVM+ and the vanilla ResNet.

## 5.2 TRAINING MODELS WITH COMPLETE PRIVILEGED INFORMATION

In this subsection, we consider the scenario where every training sample has complete privileged segmentation information. Firstly, we evaluate the performance of our proposed GoCNN on the ImageNet-0.1m dataset. Table 1 summarizes the accuracy of different models. As can be seen from the results, given the complete privileged information, our proposed GoCNN presents much better performance than compared models. The group orthogonal constraints successfully regularize the learned feature to be within the foreground and background. The trained GoCNN thus presents a stronger generalization ability. It is also interesting (although not surprising) to observe that, when foreground features with background features are combined, the performance of GoCNN can be further improved from $49.60\%$ to $50.39\%$ in terms of top-1 accuracy. One can observe that the background information indeed benefits object recognition to some extent. To further investigate the contribution of each component within GoCNN to final performance, we conduct another experiment and show the results in Table 2. In the experiments, we purposively prevent the gradient propagation from the other components except the one being investigated during training, and perform another setting on the baseline method where the background is removed and only the foreground object is reserved in each training sample, noted as *Baseline-obj*. Comparing the result of *Full GoCNN* between different classifiers, we can see that learning background features can actually improve the overall performance. And when we compare the *Fg_classifier* between *Baseline-obj*, *Only Fg* and *Full GoCNN*, we can see the importance of the background information in training more robust and richer foreground features.

Secondly, to verify the effectiveness of learning features in two different groups with our proposed method, we visualize the maximum activation value within each group of feature maps of several testing images. The feature maps are generated by the final convolution layer with $384 \times 384$ resolution input testing images. Then, the final convolution layer gives $12 \times 12$ output maps. We aggregate feature maps within the same group into one feature map by $\max$ operation. As can be seen from Figure 2, foreground and background features are well separated and the result looks just like the semantic segmentation mask. Compared with the baseline model, more neurons are activated in our proposed method in the two orthogonal spaces. This indicates that more diverse and discriminative features are learned in our framework compared with the baseline method. Finally, we further evaluate the generalization ability of our proposed method on the PASCAL VOC dataset. It is well known that an object shares many common properties with others even if they are not from the same category. A well-performing CNN model should be able to learn robust features rather than just fit the training images. In this experiment, we fine-tune different models on the PASCAL VOC images to test whether the learned features are able to generalize well to another dataset. Note that

Table 2: Validation accuracy (for 10-crop validations) of different components of GoCNN on ImageNet validation set. *Baseline-obj* refers to the baseline model trained on pure object ImageNet-0.1m dataset, *Only Bg* refers to our proposed model with foreground part gradient blocked, and *Only Fg* refers to our proposed model with background part gradient blocked. ($^*$ marks the part which shares the same classifier with the main classifier.)

|  | Top-1 Accuracy (%) | | | Top-5 Accuracy (%) | | |
|---|---|---|---|---|---|---|
|  | Main_classifier | Fg_classifier | Bg_classifier | Main_classifier | Fg_Classifier | Bg_Classifier |
| Baseline-obj | 12.45 | 12.45$^*$ | — | 24.43 | 24.43$^*$ | — |
| Only Bg | 40.36 | — | 40.36$^*$ | 67.24 | — | 67.24$^*$ |
| Only Fg | 49.15 | 49.15$^*$ | — | 73.70 | 73.70$^*$ | — |
| Full GoCNN | **50.39** | 49.60 | 40.03 | **75.00** | 74.21 | 66.98 |

Table 3: Classification results on PASCAL VOC 2012 (train/val). The performance is measured by Average Precision (AP, in %).

| Model | areo | bike | bird | boat | bottle | bus | car | cat | chair | cow | table | dog | horse | mbk | prsn | plant | sheep | sofa | train | tv | **mean** |
|---|---|---|---|---|---|---|---|---|---|---|---|---|---|---|---|---|---|---|---|---|---|
| Baseline | 95.2 | 79.3 | 90.2 | 82.8 | 52.6 | 90.9 | 78.5 | 90.2 | 62.3 | 64.9 | 64.5 | 84.2 | 81.1 | 82.0 | 91.4 | 50.0 | 78.0 | 61.1 | 92.7 | 77.5 | 77.5 |
| GoCNN | 96.1 | 81.0 | 90.8 | 85.3 | 56.0 | 92.8 | 78.9 | 91.5 | 63.6 | 69.7 | 65.1 | 84.8 | 84.0 | 83.9 | 92.3 | 52.0 | 83.9 | 64.2 | 93.8 | 78.6 | 79.4 |

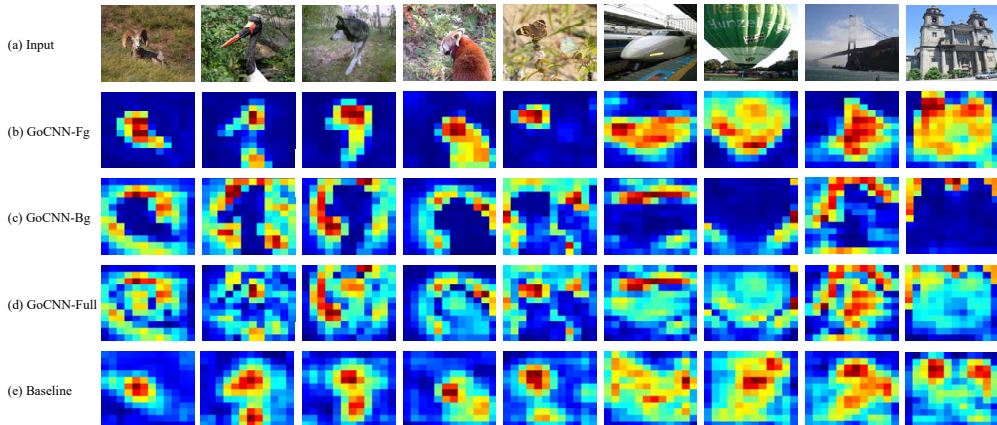

Figure 2: Activation maps of foreground feature maps (GoCNN-Fg), background feature maps (GoCNN-Bg) and all feature maps (GoCNN-Full) produced by our proposed GoCNN on ImageNet validation set. The bottom row shows the activation maps produced by the baseline model.

we add another convolution layer with a $1 \times 1$ kernel size and 512 outputs as an adaptive layer on all models. It is not necessary to add such a layer in networks without a residual structure (He et al., 2015). As can be seen from Table 3, our proposed network shows better results and higher average precision across all categories, which means our proposed GoCNN learns more representative and richer features that are easier to transfer from one domain to another.

## 5.3 TRAINING GOCNN WITH PARTIAL PRIVILEGED INFORMATION

In this subsection, we investigate the performance of different models with only using partial privileged information. The experiment is also conducted on the ImageNet-0.1m dataset. We evaluate the performance of our proposed GoCNN by varying the percentage of privileged information (*i.e.*, percentage of training images with segmentation annotations) from 20% to 100%.

The validation accuracies of GoCNN and the baseline model (*i.e.*, the ResNet-18) are shown in Table 4. From the results, one can observe that with the increasing percentage of privileged information, the accuracy will continuously increase until the percentage of privileged information reaches 80%. The performance on increasing the percentage from 40% to 100% is only 0.71% compared with 0.92% on the increasing from 20% to 40%. This is probably because the suppression losses are more effective than we expected; that is, with very little guidance from the suppression

loss, the network can already be able to separate foreground and background features and explore new features within each group.

To verify the effectiveness of GoCNN on very large training dataset with more complex CNN architectures, we conducted another experiment on the complete ImageNet-1k dataset with only 10% privileged information, and we use the 152-layer ResNet as our basic model. As can be seen from Table 5, our proposed GoCNN achieves 21.8% top-1 error while the vanilla ResNet-152 has 23.0% top-1 error. Such performance boost is consistent with the results shown in Table 4, which again confirms the effectiveness of the GoCNN.

Table 4: Validation accuracy (Top-1, in %, 1 crop validation) with 20%, 40%, 60%, 80% and 100% privileged information. Since the baseline method (ResNet-18) does not use privileged information, its validation accuracy remains the same across different tests.

| Model | 20% | 40% | 60% | 80% | 100% |
|---|---|---|---|---|---|
| Baseline (ResNet-18) | 44.26 | 44.26 | 44.26 | 44.26 | 44.26 |
| GoCNN-18 | **47.00** | **47.92** | **48.18** | **48.61** | **48.63** |

Table 5: Validation error rate (in %, 1 crop validation) with 10% privileged information on full ImageNet-1k dataset.

| Model | Top-1 err. | Top-5 err. |
|---|---|---|
| ResNet-101 He et al. (2015) | 23.6 | 7.1 |
| ResNet-152 He et al. (2015) | 23.0 | 6.7 |
| GoCNN-152 | **21.8** | **6.1** |

## 6 DISCUSSIONS

Based on our experimental results, we can also provide answers to the following two important questions.

*Does background information indeed help object recognition for deep learning methods?* Based on our experiments, we give a positive answer. Intuitively, background information may provide some "hints" for object recognition. However, though several works (Song et al., 2011; Russakovsky et al., 2012) have proven the usefulness of background information when using handcraft features, few works have studied the effectiveness of background information on deep learning methods for object recognition tasks. Based on the experimental results shown in Table 2, both the foreground classification accuracy and the overall classification accuracy can be further boosted with our proposed framework. This means that the background deep features can also provide useful information for foreground object recognition.

*Can a more precise annotation with richer information,* e.g.*, segmentation annotation, assist the image classification training process?* The answer is clearly yes. In fact, in recent years, several works have explored how object detection and segmentation can benefit each other (Dai et al., 2015; Hariharan et al., 2014). However, none of existing works has studied how image segmentation information can help train a better classification deep neural network. In this work, by treating the segmentation annotations as the privileged information, we first demonstrate a possible way to utilize segmentation annotations to assist image classification training.

## 7 CONCLUSION

We proposed a group orthogonal neural network for image classification which encourages learning more diverse feature representations. Privileged information is utilized to train the proposed GoCNN model. To the best of our knowledge, we are the first to explore how to use image segmentation as privileged information to assist CNN training for image classification.

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
