# Peer review of "Training Group Orthogonal Neural Networks with Privileged Information"

_ICLR 2017 — rejected_

[Official Review · AnonReviewer3 · rating 5 · confidence 4 · 16 Dec 2016]
**Unclear focus**

This paper proposes a modification to ConvNet training so that the feature activations before the linear classifier are divided into groups such that all pairs of features across all pairs of groups are encouraged to have low statistical correlation. Instead of discovering the groups automatically, the work proposes to use supervision, which they call privileged information, to assign features to groups in a hand-coded fashion. The developed method is applied to image classification.

Pros:
- The paper is clear and easy to follow
- The experimental results seem to show some benefit from the proposed approach

Cons:
(1) The paper proposes one core idea (group orthogonality w/ privileged information), but then introduces background feature suppression without much motivation and without careful experimentation
(2) No comparison with an ensemble
(3) Full experiments on ImageNet under the "partial privileged information" setting would be more impactful

This paper is promising and I would be willing to accept an improved version. However, the current version lacks focus and clean experiments.

First, the abstract and intro focus on the need to replace ensembles with a single model that has diverse (ensemble like) features. The hope is that such a model will have the same boost in accuracy, while requiring fewer FLOPs and less memory. Based on this introduction, I expect the rest of the paper to focus on this point. But it does not; there are no experimental results on ensembles and no experimental evidence that the proposed approach in able to avoid the speed and memory cost of ensembles while also retaining the accuracy benefit.

Second, the technical contribution of the paper is presented as group orthogonality (GO). However, in Sec 4.1 the idea of background feature suppression is introduced. While some motivation for it is given, the motivation does not tie into GO. GO does not require bg suppression and the introduction of it seems ad hoc. Moreover, the experiments never decouple GO and bg suppression, so we are unable to understand how GO works on its own. This is a critical experimental flaw in my reading.

Minor suggestions / comments:
- The equation in definition 2 has an incorrect normalizing factor (1/c^(k)^2)
- Figure 1 seems to have incorrect mask placements. The top mask is one that will mask out the background and only allow the fg to pass

[Official Review · AnonReviewer1 · rating 6 · confidence 4 · 16 Dec 2016]
**No Title**

The starting point of this work is the understanding that by having decorrelated neurons (e.g. neurons that only fire on background, or only on foreground regions) one provides independent pieces of information to the subsequent decisions. As such one gives "complementary viewpoints" of the input to the subsequent layers, which can be thought of as performing ensembling/expert combination within the model, rather than using an ensemble of networks. 

For this, the authors propose a sensible method to decorrelate the activations of intermediate neurons, with the aim of delivering complementary inputs to the final classification layers: they split intermediate neurons to a "foreground" and a "background" subset, and append side-losses that force them to be zero on background and foreground pixels respectively. 

They demonstrate that this can improve classification on a mid-scale classification example (a fraction of imagenet, and a ResNet with 18, rather than 150 layers), when compared to a "vanilla" baseline that does not use these losses.

I enjoyed reading the paper because the idea is simple, smart, and seems to be effective. 
But there are a few concerns;
-firstly, the way of doing this seems very particular to vision. In vision one knows that masking the features (during both training and testing) helps, e.g.

[Official Review · AnonReviewer2 · rating 6 · confidence 4 · 19 Dec 2016 (modified: 30 Jan 2017)]
**No Title**

This paper proposes to learn groups of orthogonal features in a convnet by penalizing correlation among features in each group.  The technique is applied in the setting of image classification with “privileged information” in the form of foreground segmentation masks, where the model is trained to learn orthogonal groups of foreground and background features using the correlation penalty and an additional “background suppression” term.


Pros:

Proposes a “group-wise model diversity” loss term which is novel, to my knowledge.

The use of foreground segmentation masks to improve image classification is also novel.

The method is evaluated on two standard and relatively large-scale vision datasets: ImageNet and PASCAL VOC 2012.


Cons:

The evaluation is lacking.  There should be a baseline that leaves out the background suppression term, so readers know how much that term is contributing to the performance vs. the group orthogonal term.  The use of the background suppression term is also confusing to me -- it seems redundant, as the group orthogonality term should already serve to suppress the use of background features by the foreground feature extractor.

It would be nice to see the results with “Incomplete Privileged Information” on the full ImageNet dataset (rather than just 10% of it) with the privileged information included for the 10% of images where it’s available.  This would verify that the method and use of segmentation masks remains useful even in the regime of more labeled classification data.

The presentation overall is a bit confusing and difficult to follow, for me.  For example, Section 4.2 is titled “A Unified Architecture: GoCNN”, yet it is not an overview of the method as a whole, but a list of specific implementation details (even the very first sentence).

Minor: calling eq 3 a “regression loss” and writing “||0 - x||” rather than just “||x||” is not necessary and makes understanding more difficult -- I’ve never seen a norm regularization term written this way or described as a “regression to 0”.

Minor: in fig. 1 I think the FG and BG suppression labels are swapped: e.g., the “suppress foreground” mask has 1s in the FG and 0s in the BG (which would suppress the BG, not the FG).


An additional question: why are the results in Table 4 with 100% privileged information different from those in Table 1-2?  Are these not the same setting?

The ideas presented in this paper are novel and show some promise, but are currently not sufficiently ablated for readers to understand what aspects of the method are important.  Besides additional experiments, the paper could also use some reorganization and revision for clarity.

===============

Edit (1/29/17): after considering the latest revisions -- particularly the full ImageNet evaluation results reported in Table 5 demonstrating that the background segmentation 'privileged information' is beneficial even with the full labeled ImageNet dataset -- I've upgraded my rating from 4 to 6.

(I'll reiterate a very minor point about Figure 1 though: I still think the "0" and "1" labels in the top part of the figures should be swapped to match the other labels.  e.g., the topmost path in figure 1a, with the text "suppress foreground", currently has 0 in the background and 1 in the foreground, when one would want the reverse of this to suppress the foreground.)

[Author Response · Yunpeng Chen · 21 Jan 2017]
**Update**

We've already updated the paper. 
- The abstract and introduction have been rewritten with more explanation (on the motivation) and comparison. 
- The difference from ensemble models was highlighted in the related works.
- We found that the Fig 1. is a bit confusing and have already updated it in the revised revision.
- Eqn 3. has been corrected.
- New results on ImageNet dataset.

[Final Decision · Program Chairs · 06 Feb 2017]
**ICLR committee final decision**

This paper was reviewed by three experts. While they find interesting ideas in the manuscript, all three point to deficiencies (lack of clean experiments, clarity in the manuscript, etc) and recommend rejection. I believe there are promising ideas here, and this manuscript will be stronger for a future deadline.